# Cooperative Learning Promoting Cultural Diversity and Individual Accountability: A Systematic Review

**Tong Zhou [1] and Jordi Colomer [2],***

[1]  Department of Physical Education, Korea University, Seoul 02841, Republic of Korea; zhou941002@korea.ac.kr
[2]  Department of Physics, University of Girona, 17003 Girona, Spain
*   Correspondence: jordi.colomer@udg.edu

**Abstract:** Cooperative learning in physical education not only promotes the overall development of students, but also lays the foundation for lifelong learning and sustainable development from the perspectives of cultural integration and social responsibility. This study examined students' physical, social, emotional, and cognitive outcomes on the one hand. On the other hand, it focuses on the potential of cooperative learning to promote cultural diversity and personal responsibility. A systematic literature review of 50 articles selected according to the PRISMA guidelines revealed that the focus of CL applications varied according to the age of the students and multicultural contexts. At the micro level, physical and social domains were emphasized, while cognitive domains received less attention. CL was found to benefit motor skills (coordination, flexibility, strength) in students of different ages, to reduce negative emotions while promoting social skills and teamwork, and to improve cognitive skills and memory in junior students. At the macro level, CL can help students to improve self-reflection, reduce negative behaviors, and increase personal responsibility and cultural integration. The focus of researchers from different countries reflects educational philosophies and cultural differences, increasing the flexibility and universality of CL.

**Keywords:** cooperative learning; motor skills; cultural diversity; individual accountability; physical education

## 1. Introduction

Physical education (PE) is no longer just a means of promoting students' physical health, but also plays an important role in developing students' social, teamwork, and leadership skills, and personal responsibility as well [1,2]. In the context of promoting the goal of education for sustainable development (ESD) in the 21st century, the cooperative learning (CL) approach has attracted attention as a teaching strategy that can effectively contribute to the realization of this goal through group work [3]. While the number of school and university educational programs devoted to promoting sustainable education are still sparse and disconnected to set global objectives, the development of competences for sustainable development have received much attention [4]. Cooperative learning is considered an instructional approach that promotes cooperative skills and directs the basics of the development of students' sustainable goals through the development of social, cognitive, physical, and affective aspects [5,6].

The continued exploration of CL instructional approaches has moved beyond the limitations of traditional approaches that focus solely on outcomes and achievement [7]. CL focuses on the learning process, shifting from the old 'teacher-centered' model to a 'student-centered' model [8]. Through continuous collaboration and task allocation between students, CL aims to enable students to become autonomous and responsible learners [9]. Through cooperative learning groups, culturally diverse students have the opportunity to enhance interpersonal teamwork, thereby improving social skills [10]. The process of working together helps to undermine cultural barriers, improve understanding

and communication between students from diverse cultural backgrounds and promote cultural integration [11]. In particular, a quality education that promotes gender equality and reduces inequalities is primarily dependent on educational institutions providing sustainable educational approaches, values, and competences for sustainable development.

One way to characterize cooperative learning is as an ESD approach where students cooperate in small, diverse (heterogeneous) groups to maximize both their own and others' learning [12]. Cooperative learning promotes students to work together to achieve common goals, increases positive interdependence, and enhances face-to-face communication and social and group interaction, among other abilities. Students that participate in cooperative learning work together to discover and resolve problems, exchange ideas, or carry out a project or a challenge [13]. It addresses strategic action by utilizing communication and deliberative abilities, as well as peer-to-peer engagement, which enhances the acquisition of long-term knowledge and skills.

In addition, CL promotes students to develop peer-to-peer promotive interaction and individual accountability that enhances students' enthusiasm for learning, thus better developing motor skills that increase students' positive interdependence. Also, CL has a positive impact on mutual respect between pupils, positive emotions and attitude formation. This, coupled with individual reflection and discussion between member groups, stimulates collisions in thinking and promotes sharing of knowledge while improving learning efficiency. Therefore, CL works under the principles of leaving no one behind and promoting cultural diversity through equity and inclusivity, while developing competences for sustainable development [14]. One may argue that CL offers numerous beneficial advantages for the growth of students' social, group, and individual aspects.

Previous systematic reviews have found that more studies have addressed social skills, teacher–student relationships, and motivation to learn [15,16]. Nevertheless, the majority of reviews did not quantitatively examine the impact of CL on students' multiple learning outcomes (motor skills, social skills, motivation for self-directed learning, affective experiences, etc.), nor did they analyze how the CL was applied in diverse age groups and multicultural contexts. Also, we can argue that most reviews did not explore the role of CL in developing global awareness and civic responsibility in the context of the current global goals of education for sustainable development. Therefore, there is a need to conduct a systematic review of research on CL in PE, that focuses on examining the physical, social, effective, and cognitive areas of student PE outcomes along with the promotion of cultural diversity within CL dimensions.

## 2. Literature Review

### 2.1. Cooperative Learning

CL is a way of learning about and implementing PE that leads to improvements in both teaching and learning, encouraging students to work together in small, structured, and heterogeneous groups to achieve shared learning goals [17]. In PE, CL is widely acknowledged as an effective method to promote students' social, physical, cognitive, and emotional development [18]. This instructional approach emphasizes the organic integration of individual students with small group dynamics, fostering cooperative efforts among group members to collectively construct knowledge, share resources, and mutually enhance the learning process [19].

In practical terms, CL manifests in various forms within PE, including group discussions, collaborative projects, and peer teaching [20,21]. These formats aim to cultivate students' critical thinking, teamwork skills, and effective communication abilities [22,23]. Through close interaction among group members, students have the opportunity to achieve a comprehensive learning experience in cooperation and teamwork. This not only facilitates individual progress but also elevates overall comprehensive literacy, fostering intrinsic motivation among students [24,25].

Dyson and Grineski [26] presented five elements and five structures that physical educators can use to achieve the national standards (National Association for Sport and

Physical Education [NASPE] 1995), particularly standards five, six, and seven, which emphasize social interaction, inclusion, acceptance of others, and the development of cognitive skills. Moreover, CL has been widely applied across different educational stages, including primary schools, secondary schools, and universities [27–29]. Dyson [30] found that CL enabled primary school students with diverse developmental levels to enhance motor skills, develop social skills, take responsibility for their own progress, and assist others in skill improvement.

### 2.2. Cooperative Learning and Social Constructionism

CL has garnered widespread attention and research in the general education field, with social constructivism providing a robust theoretical foundation for understanding its effectiveness. Studies indicate that through promotive interaction, students can gain a deeper understanding and application of knowledge, aligning with the social nature of learning emphasized by social constructivist theory. Interactions and cooperation within members' groups not only contribute to knowledge construction but also cultivate students' critical thinking, promote communication skills, and enhance problem-solving abilities. This research underscores the significance of CL in fostering an enriched educational experience, aligning with the principles of social constructivism in the realm of general education [31].

Additionally, CL contributes to alleviating students' feelings of anxiety and enhances their motivation for learning [20,32,33]. Through CL with peers, students find it easier to develop an interest in learning, and sharing experiences within groups helps alleviate individual stress during the learning process [34]. Therefore, by integrating social constructivist theory with CL, educators can better design instructional activities to enhance students' academic achievements and overall development [35].

Particularly, in PE, CL is usually directed to the application of social constructivist basis which emphasizes the development of positive teamwork and social skills [36]. The application of CL in the field of developing psychomotor activities within students not only promotes positive relationships between peers, but also improves individual learning. In team sports, CL helps students to better understand each other's roles and responsibilities [20], so that they can effectively avoid the phenomenon of hitchhiking in cooperation. However, this phenomenon can be understood as the implementation of social constructivism in PE that needs to take into account specific factors such as individual differences and ability levels [37]. This requires teachers to know enough about their pupils to be able to group them appropriately. Therefore, when integrating social constructivism theory into PE, it is important to avoid both the disagreements that arise from being in the same group for long periods of time and the uneven distribution of group tasks that can lead to undermining cultural differences.

### 2.3. Cooperative Learning for Physical Education Students

Research has shown that the CL instructional approach has been used in physical education and has been found to be effective in promoting the following instructional outcomes: physical, effective, social, and cognitive [18]. Research has also found that CL is effective in promoting student motivation and is an effective teaching strategy to develop the basic psychological needs postulated in self-determination theory [38]. The effects of CL have been shown to be positive in various sports disciplines such as basketball, football, and handball [39–41].

### 3. Method

A systematic evaluation was conducted to assess the current status of CL in PE. The flowchart is a visual representation of the study selection process to conduct the current systematic review in compliance with the PRISMA guidelines [42], including the number of records identified, screened, assessed for eligibility, and included in the review. The reliability and validity of the screening process and results were also analyzed to improve the

scientific validity of the literature selection. This systematic review protocol has been registered on INPLASY (https://inplasy.com/inplasy-2024-5-0096/, accessed on 28 April 2024). The registration number is INPLASY202450096. This protocol was performed in accordance with the preferred reporting items for systematic reviews and meta-analysis protocol. Ethical approval is unnecessary because this is a literature-based study.

### 3.1. Data Collection

Relevant literature was retrieved by searching the core databases of Web of Science (WOS) and SCOPUS. Both databases are widely recognized as the most authoritative and reputable publishers' databases [43,44]. To avoid any omissions, a snowballing strategy was carried out at the same time [45].

### 3.2. Search Limits

Based on systematic review and meta-analysis guidelines [42], the following PICO strategies were included: participants (e.g., adolescents, high school students, college students, children, kids), interventions (e.g., curricula, training, physical education, cooperative), types of study (quantitative, mixed studies that included quantitative studies), comparative subjects (e.g., 'physical education,' 'cooperative learning'), and outcomes (e.g., cognitive, social, affective, motor). The study set the time span of database search from January 2000 to 31 December 2023 to better fit the continuity and completeness of the study made in the last 20 years. The inclusion and deletion criteria based on the research objectives are shown in Table 1.

**Table 1.** Literature search list.

| Inclusion Criteria | Exclusion Criteria |
|---|---|
| 1. Research primarily focused on CL. | 1. Research sourced from conference proceedings, books, magazines, news, and posters. |
| 2. Studies must report on research involving CL in PE. | 2. Research unrelated to learning outcomes (including duplicate articles). |
| 3. These studies should be published in peer-reviewed journals. | 3. The results of the experiment did not report the impact of CL on learning outcomes (any of the four domains). |
| 4. Research articles must be in English (due to language constraints, English is the main language of choice). | Exclusion criteria. |
| 5. Quantitative studies and mixed studies that include quantitative studies on CL. | 1. Research sourced from conference proceedings, books, magazines, news, and posters. |
| 6. Full-text availability is required. | 2. Research unrelated to learning outcomes (including duplicate articles). |

### 3.3. Data Analysis

Screening resulted in 209 articles and 32 were collected by snowballing, for a total of 241 articles. Based on the principle of censoring, 166 duplicates were eliminated, and 50 articles remained after manual screening through content analysis, eliminating articles that did not meet the topic relevance, were not accessible in full text, and in order to further validate the learning effects of CL on students, studies in which non-students were the subjects of the study were also excluded, as well as articles with non-quantitative studies (quantitative-only studies and mixed studies that included quantitative studies); this reduction was achieved through manual screening, whereby two people independently screened and compared the articles to ensure that the study was reliable [46].

Agreement between the two reviewers was assessed using inter-rater reliability calculations widely used in educational statistics and measurement. The results showed perfect agreement, with both reviewers selecting the same 47 articles from a total of 241 through

independent screening and including three through discussion, with the final 50 documents included in the analysis, as shown in Figure 1.

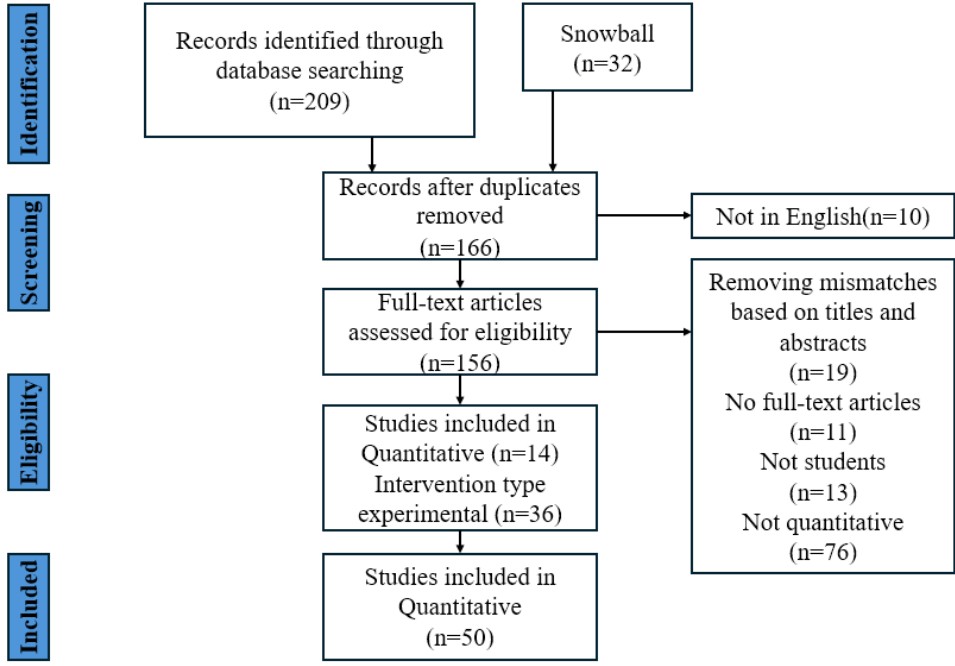

**Figure 1.** Data-cleaning flowchart.

The inter-rater reliability (agreement) was calculated as follows:

$$K \text{ value (average inter-agreement) } k = M/N \times 100\%$$

$$\text{Reliability (R) } R = (n \times k)/(1 + (n - 1) \times k)$$

The K value is 94%, and the reliability (R) is 0.9987, which is greater than 0.9. This indicates that the agreement between the two reviewers is excellent, as a reliability value above 0.9 is considered highly reliable [46].

## 4. Results

### 4.1. Cooperative Learning Integration Effects

A specific review and examination culminated in a specific review of the impact of CL interventions under study in different physical education settings (see Table S1). The impact of CL on students' outcomes was explored from different perspectives. And the systematic review found that 13 (26%) articles were cross-sectional studies conducted by distributing questionnaires and the remaining 37 (74%) articles were experimental studies conducted by collaborative learning interventions. The study population ranged from a minimum of 3 to a maximum of 1332.

#### 4.1.1. Impact of Age and Education Level

Intervention studies on the CL in PE have covered students of different age levels, from kid to high school [47,48], and even college [49]. This reflects the flexibility and universal applicability of the CL, and this became a common and effective teaching method.

A systematic review of the results revealed that CL has been applied to children (2, 4%), elementary schools (13, 26%), junior schools (19, 38%), high schools (4, 8%), and college (12, 24%), as shown in Figure 2. CL has been applied to fewer studies with children and most of the studies were conducted with junior school teaching.

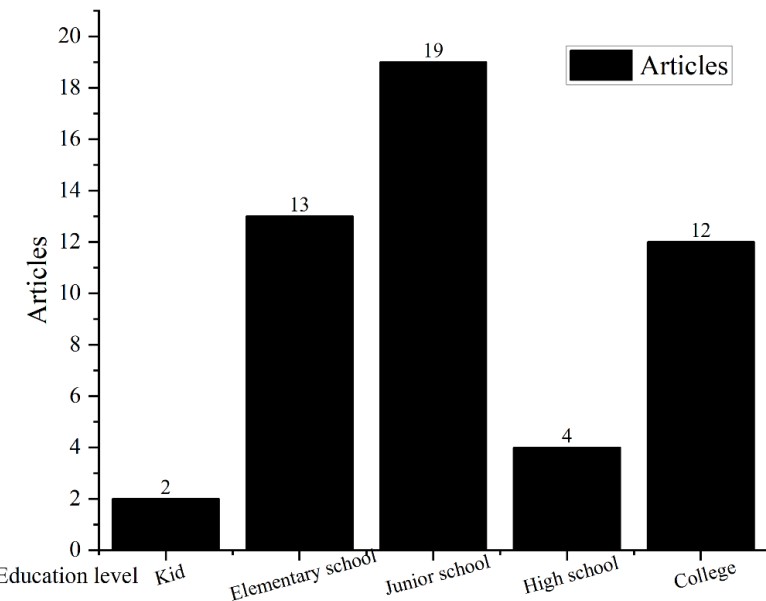

**Figure 2.** Educational level of CL intervention recipients.

The intervention of CL plays a crucial role in the comprehensive development of students' social skills across different stages, see Table 2. Research on emotionalization in children is a key concern addressed by CL. Interventions through CL scenarios targeting social interaction in children with disorders have been found to effectively reduce the ratio of inappropriate interactions [50]. Furthermore, CL positively contributes to improving children's negative psychological states, fostering their psychological and moral development [51].

**Table 2.** Education level of main research domains and content.

| Education Level | Articles | Main Research Domains and Content |
| --- | --- | --- |
| Kid | 2 | Effects: interventions for negative emotions and social disorders |
| Elementary | 13 | Physical: development of a motor skill; Social: development of social skills; Effective: attitudes towards learning, responsibility, emotional learning; Cognitive: cognitive learning and memory skills, critical thinking skills |
| Junior | 19 | Physical: development of a particular motor skill; Effective: perceptual skills (motivation), attitude development, personal responsibility; Social: social skills |
| High | 4 | Physical: development of a particular motor skill; Effective: perceived competence (motivation), disruptive behavior, participation Social: social competence; Emotional: self-esteem, perceived competence; Cognitive: development of cognitive competence; |
| College | 12 | Social: social skills; Effective: self-esteem, perceptual skills; Cognitive: cognitive development; |

During the primary school stage, CL, by promoting collaboration and interaction among students, effectively nurtures social skills and collaborative abilities. Altınkök's [52] study revealed significant improvements in students' motor skills through CL interventions in first-grade students in Turkey, with noticeable impacts. Cooperation among groups not only enhances collective improvement in games and physical activities but also strengthens interaction within the groups. Goudas and Magotsiou's [38] study similarly found that cooperative physical education courses have positive effects on students' social skills and attitudes toward group work.

In junior and high school stages, CL not only focuses on the collective construction of PE knowledge, but also emphasizes the cultivation of social skills. For example,

Şahin [53] demonstrated that CL situations significantly influence students' academic performance compared to traditional teacher-centered teaching methods. Additionally, Iserbyt et al.'s [54] research indicated that, in the high school stage in Belgium, CL can enhance students' social skills and reduce gender inequality through role reversal and definition.

In recent years, the application of CL has become increasingly prevalent in universities and pre-service education. The enhancement of social capabilities is crucial for university students preparing to enter society [55]. Ortuondo Bárcena et al. [56] conducted a four-month intervention, revealing significant changes in students' social abilities through pre- and post-experimental comparisons. Moreover, Cohen and Zach [57] demonstrated that establishing a "learner community" effectively enhances students' and teachers' teaching efficacy and planning skills. This suggests that CL holds significant value in training prospective physical education teachers.

### 4.1.2. National and Cultural Differences

National and cultural differences offer us a crucial perspective on the impact of the application of CL instructional approaches in various cultural contexts. Analyzing the results from Figure 3, Spain (19), France (8), China (4), Turkey (3), and Israel (3) emerge as the top five countries with the highest number of published articles related to cooperative learning. Upon specifically examining the content of publications from these countries, diverse emphases and focus areas become apparent.

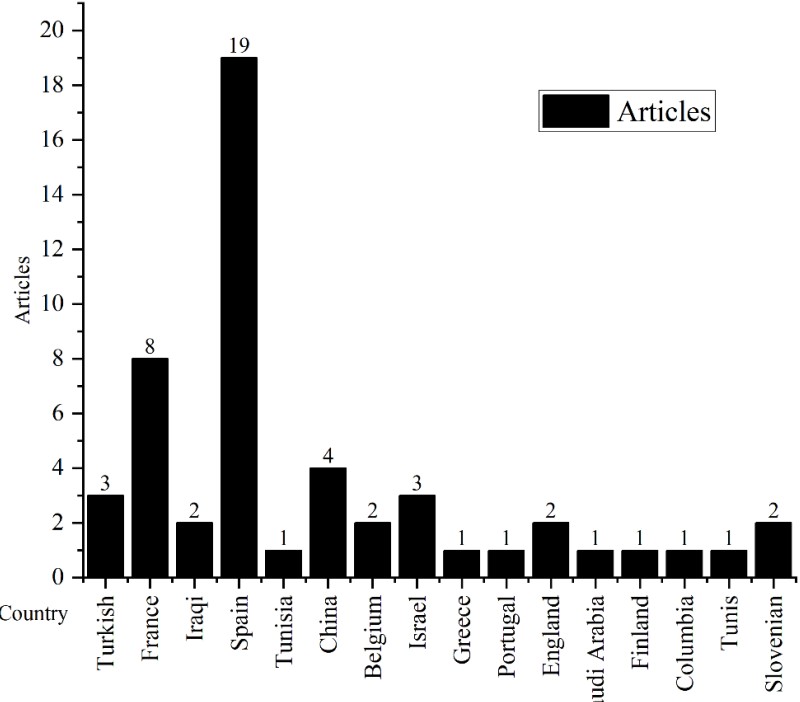

**Figure 3.** Number of regional cultural publications.

Spanish research predominantly emphasizes subject collaboration, social objectives, intrinsic motivation, and physical fitness. Conversely, French studies place more emphasis on student acceptance and gender equality, reflecting a greater concern for collectivist values in France. These cultural differences manifest prominently in the field of CL, providing a diverse perspective for the expansion and applicability of CL in various educational environments.

Through review of the literature, the results in Table 3 show the importance of CL in promoting the development of cultural diversity, cross-cultural communication, global awareness, critical thinking, communication skills, social integration, and civic responsibility.

**Table 3.** The facilitating role of cooperative learning.

| Factors | CL |
|---|---|
| *Cultural diversity* | Improving understanding and respect for different cultures by working with colleagues. |
| *Cross-cultural communication* | Creating opportunities for cultural exchange, reducing prejudice, and promoting tolerance and open-mindedness. |
| *Global awareness* | Understanding the customs and cultures of different countries and regions and promoting global awareness and citizenship. |
| *Critical thinking* | To work together to explore the reasons behind different cultural phenomena and to develop critical thinking skills. |
| *Communication skills* | Collaborating with people. |
| *Social integration* | To promote understanding and integration between different cultural groups through cooperation. |
| *Civic responsibility* | To focus on the social problems faced by different cultural groups and to develop a sense of civic responsibility. |

*4.2. Cognitive Development*

The use of CL in physical education can effectively enhance students' cognitive learning achievement and have a significant positive effect on promoting students' cognitive development [53]. Rakha [58], amidst the COVID-19 period, organized online lectures through a learning management system, revealing that students who experienced CL achieved higher average grades and scores in intelligence tests. Comparative to direct instruction, students' cognitive learning abilities and memory capacities were significantly improved after undergoing six months of CL interventions [59].

*4.3. Improvement of Motor Skills*

One of the core objectives of PE is to foster the development of students' motor skills and physical fitness. Research indicates that CL has a positive impact on students' coordination, flexibility, and strength, among other aspects [52]. This suggests that CL contributes to shaping students' motor skills, enabling them to perform better in physical activities. The study further emphasizes the necessity of prolonged use of CL scenarios for the development of students' motor skills.

Moreover, this review reveals that CL scenarios have been confirmed in sports such as soccer, basketball, handball, track and field, volleyball, and swimming, effectively promoting the improvement of motor skills [41,48,60]. CL not only facilitates the development of motor skills, but Barrett's [19] research also found that, through CL scenario instruction, the performance of low-skilled students is comparable to that of moderate- and high-skilled students. However, whether different sports projects yield similar effects requires further validation.

*4.4. Social Skills and Team Collaboration*

In PE, CL goes beyond its benefits for energy expenditure in sports; it also underscores the significance of social skills and teamwork. Research by Baena-Morales, Jerez-Mayorga, Fernández-González, and López-Morales [49] indicates that CL is considered an effective technique for developing social relationships and capabilities, praising it as a sustainable instructional approach. Through CL, students can acquire effective communication within a team, actively support one another, and coordinate task assignments, laying a solid foundation for future team sports activities and careers. However, findings by Vega-Ramírez, Vidaci, and Hederich-Martínez [55] suggest that while this collaboration can indeed enhance social interaction and social skills, it may be influenced by external factors such as family, work, and other social commitments.

*4.5. Enhancement of Intrinsic Motivation and Exercise Intent*

Student intrinsic motivation is a crucial factor in fostering active participation in physical activities. CL plays a role in promoting students' intrinsic motivation, making them more willing to actively engage in various sports activities and enhancing both their intrinsic motivation and exercise intent [61]. When compared with traditional teaching methods, the creation of a motivational climate within a CL environment significantly improves students' task involvement and self-engagement [62]. Additionally, Garví-Medrano, et al. [63] substantiate that a five-month CL intervention plan can significantly enhance students' willingness to engage in physical activities. By fostering a CL atmosphere, students become more enthusiastic about participating in PE classes, cultivating a higher level of intrinsic motivation.

Studies have also found that the impact of CL and direct teaching models on the motivational climate in PE classrooms varies. Regardless of the class's gender composition or the teaching model employed, students perceive mastery climate as superior to performance climate [64]. It is emphasized that motivational climate cannot be solely attributed to teaching models, as their effective implementation requires experienced teachers. Extensive experience enables professors to actively cultivate a positive learning atmosphere, and CL imposes certain demands on teachers' capabilities. Trabelsi, et al. [65] discovered that a video-based peer feedback approach effectively enhances student engagement in the learning process, especially among female students.

*4.6. Cultivation of Emotional States and Emotional Intelligence*

In PE under CL, emphasis is not only placed on the aspects mentioned earlier but also on the crucial outcomes related to emotions. The emotional development of students can help them attain emotional support, aligned with the basic psychological needs theory [63]. Palau-Pamies and Tortosa-Martínez [66] conducted an analysis of the impact of CL on students' basic psychological needs compared to traditional models. After validating using the basic psychological needs scale, they found statistically significant improvements in autonomy, competence, and social relationships under CL. Moreover, the positive atmosphere created by CL effectively enhances students' satisfaction with their basic psychological needs.

Following prolonged exposure to CL, students experience positive perceptions such as cooperation, connection, enjoyment, novelty, as well as negative perceptions like disappointment. Therefore, the implementation in PE classes should consider both positive and negative thoughts of students [24]. However, negative emotions can be effectively improved through CL interventions, reducing inappropriate actions and negative emotions [51]. It is worth noting that this research primarily focused on children, and whether similar reductions in negative emotions can be achieved in other age groups remains uncertain. An intriguing study discovered a significant correlation between CL and emotional intelligence. The application of CL in PE can guide students toward adaptive motivation and the development of emotional intelligence [67].

*4.7. Global Awareness and Civic Responsibility*

It is noteworthy that interdisciplinary educational approaches promote sustainable development in elementary education [68]. According to the United Nations Sustainable Development Goals, education is crucial for students to create value, specifically Sustainable Development Goal 4 (Quality Education) and Sustainable Development Goal 5 (Gender Equality). CL, as an effective technique for sustaining social relationships and capabilities, can help students collaborate and learn across different regions [49]. By introducing students to CL, promoting a higher level of understanding, reflection, and critical thinking development, students gain a deeper insight into global sustainable education development [68]. As members of the global community, CL not only effectively enhances prosocial behaviors and reduces disruptive behaviors [69], but also diminishes gender inequality [70]. The key to this awareness is critical thinking, and CL environments contribute to creating a learning setting conducive to critical thinking in the context of PE [29].

Facing the continuous development of global informatization, computer and internet technologies actively support students in adopting positive attitudes toward global information and communication technologies. Through CL, students can effectively enhance their critical awareness of evaluating internet health information [71]. This critical awareness provides a favorable environment for CL to develop students' intercultural understanding and respect for diverse cultures [72]. By working with peers of different nationalities and cultural backgrounds, students can enhance their understanding of different values and customs, thereby increasing inclusiveness and openness [73]. For example, in PE classes, students from different countries can work together to demonstrate traditional sports of their respective nationalities and learn about the importance and value of sport in different nationalities. This kind of cultural exchange helps to break down prejudices and develop an attitude of mutual respect.

CL also allows pupils to participate in solving real problems and to develop a sense of citizenship and social responsibility [74,75]. Teachers can design cooperative projects in which teams of students focus on community issues or environmental protection. Through the process of collaborative research, discussion and resolution, students not only learn teamwork, but also demonstrate their concern for social issues and their sense of participation [76].

## 5. Discussion

The results of this study, by analyzing 50 pieces of literature, showed that the main focus of research differs for students of different ages. This is mainly related to the growth characteristics and cognitive development of the students. There is a relative lack of research on children, with most research focusing on secondary schools and universities.

### 5.1. Collaborative Learning on Cognitive, Emotional, Social, and Motor Learning

CL not only contributes to students' cognitive development, but also has positive motor, social, and emotional effects. At the cognitive level, CL helps students to construct knowledge. The process of division of labor and mutual problem solving within the group can deepen students' understanding of competence development and develop critical thinking and metacognitive skills, which enhances students' cognitive abilities as shown in [43,59]. At the same time, the multiple perspectives of different students help to expand ideas and stimulate innovative thinking. In terms of motor skill development, members of CL groups can observe and coach each other, which is conducive to the transfer and internalization of motor skills. The cooperative atmosphere can also increase the interest and participation of learners, thus achieving higher results with less effort. In physical education, the use of CL strategies often results in better motor skill development. In addition, CL helps to develop students' social skills, such as communication and teamwork. Group work requires students to exchange ideas and coordinate the division of labor with others, therefore they are challenged to learn to think differently and to respect differences, which contributes to the development of students' interpersonal skills and social adaptability. Finally, CL also affects students' emotional experience. Compared to competition, a cooperative environment can reduce anxiety and increase the sense of achievement; the collective identity of the group can also enhance students' self-efficacy and cultivate positive emotions, all of which are consistent with the findings of Fernandez-Rio and Zhang [24,51].

### 5.2. Multiple Developmental Benefits of Cooperative Learning

Cooperative learning as a teaching strategy not only promotes mutual learning between students of different levels, but also promotes intercultural learning in multicultural groups [77]. First, cooperative learning provides opportunities for students to interact with peers while meeting their learning needs [78]. In the application of CL, forming heterogeneous groups allows students to collaborate and learn from each other, thereby better addressing individual differences [29]. At the same time, CL plays an important role in promoting the development of cultural diversity, cross-cultural communication, global awareness, critical thinking, communication skills, social integration, and civic responsi-

bility. Heterogeneous group work not only helps students to recognize and experience different approaches to culture and promote mutual understanding, but also cultivates open and tolerant attitudes and improves intercultural communication skills [73]. In the process of discussing and completing tasks together, group members need to listen to each other's different perspectives, stimulate critical thinking and improve communication and cooperation skills [29]. In addition, the atmosphere of group work is also conducive to enhancing students' sense of social responsibility and citizenship [54]. Therefore, the creation of mixed-ability groups in culture classes can promote the development of multicultural understanding, critical thinking, and social responsibility among students.

CL indeed proves effective in promoting students' physical, cognitive, social skills, and affective, aligning with the findings of previous studies [14]. However, the outcomes of cooperative learning should also include the global awareness, cultural sensitivity, and civic responsibility found in this study, which are needed for the development of sustainable education. In this process, cooperative learning has great potential for development. In addition, the use of web-based technological tools can provide a rich collaborative environment for students and improve the efficiency and quality of group learning [79]. Technology provides a new classroom environment for collaborative learning. In addition, in cooperative learning, students use online platforms and applications to co-develop different learning resources and tasks, and students can facilitate mutual learning and reflection through online comments and interactions [80–82]. CL in this technological extension better meets students' learning interests and intrinsic psychological needs.

Additionally, CL actively encourages students to take on leadership roles, promoting not only the development of individual leadership but also fostering a sense of responsibility [54]. The Global Goals for Sustainable Development require students not only to be proactive leaders, but also to become more aware of their responsibilities and to enhance pro-social behavior in order to contribute to the sustainable development of their societies [83]. CL also highlights each student's specific strengths.

This strength, through cooperative learning, provides more balance and possibilities for teaching and learning in physical education. Based on deep understanding and balance, cooperative learning takes into account the needs of individual differences as much as possible, while maintaining fairness and effectiveness in teaching and learning. This is particularly significant for enhancing the enthusiasm of female students in participating in physical activities, consistent with the findings of [70]. CL can effectively promote educational equity, ensuring the achievement of sustainable development goals in education.

### 5.3. Potential Challenges of Cooperative Learning

While the integration of personalized education with CL presents new opportunities for education, educators must simultaneously consider and address potential issues and challenges [84,85]. In CL, group composition may lead to significant disparities in academic abilities, resulting in students being unable to fully participate in the cooperation [86,87].

Second, with increasing globalization, the need for communication between students from different geographical backgrounds and the use of technological tools can exacerbate the problem of the digital divide [88], as not all students have effective access to and use of digital tools. While geographically diverse cultures are conducive to fostering multicultural awareness, they can also cause friction and require cooperative learning to develop students' critical thinking and inclusiveness [89]. As a result, students may feel peer pressure in small groups or be marginalized due to social factors, which ultimately affects the individual learning experience.

Last, the assessment of cooperative learning poses one of the challenges in using the cooperative learning model. Traditional assessment methods struggle to reflect individual contributions in cooperative learning, necessitating more comprehensive evaluation and assessment methods [85].

## 6. Conclusions

This study, examining the performance outcomes of students in PE under the context of CL, encompasses the enhancement of students' motor skills, cultivation of social skills and teamwork, elevation of autonomous and motivational factors in PE, development of emotional states and emotional intelligence, promotion of global awareness and cultural sensitivity, and a specific examination of civic responsibility.

CL demonstrates significant positive impacts in PE, effectively improving students' cognitive development and academic performance while increasing enthusiasm for physical participation. On the other hand, CL efficiently eliminates negative emotions and enhances students' critical thinking. Moreover, in social construction, CL effectively promotes students' individual social skills and interpersonal abilities. In the process of sustainable development, CL scenarios encourage students to heighten their personal sense of responsibility, fostering positive societal actions and promoting gender equality in educational opportunities. This implies that CL can effectively advance students' development in various aspects, providing a fresh perspective for sustainable educational development. In the examination of national and cultural differences, diverse emphases in CL research among different countries reflect variations in educational philosophies and cultural values. This diverse environment enhances the flexibility and adaptability of cooperative learning, facilitating their application in various cultural contexts.

The study also found that the implementation of cooperative learning in physical education programs can not only improve students' motor skills, but also develop their social skills, teamwork, autonomy, and motivation, promote the development of emotional intelligence, enhance global awareness and cultural sensitivity, and strengthen civic awareness. This suggests that the rational use of cooperative learning strategies in physical education programs can promote the all-round development of students. Therefore, physical education teachers should actively explore ways to effectively implement cooperative learning in practice, create a good cooperative environment, guide students to actively participate in cooperative activities, make full use of the advantages of cooperative learning, and promote the coordinated development of students in physical, intellectual, social, and emotional aspects.

However, this study acknowledges certain limitations due to the inclusion of specific literature. On one hand, factors such as sample selection, differences in experimental designs, and the use of measurement tools may impact research outcomes. On the other hand, the scope and depth of studies might be limited, preventing a complete revelation of unexpected situations in CL within PE. Future research needs to look more closely at how physical education teachers can effectively implement CL interventions in PE to support pupils' overall development.

**Supplementary Materials:** The following supporting information can be downloaded at: https://www.mdpi.com/article/10.3390/educsci14060567/s1. Table S1: Cooperative Learning.

**Author Contributions:** Conceptualization, T.Z. and J.C.; methodology, T.Z.; software, T.Z.; validation, T.Z. and J.C.; formal analysis, T.Z.; writing—original draft preparation, T.Z. and J.C.; writing—review and editing, T.Z. and J.C.; supervision, J.C. All authors have read and agreed to the published version of the manuscript.

**Funding:** This research received no external funding.

**Data Availability Statement:** No new data were created or analyzed in this study. Data sharing is not applicable to this article.

**Conflicts of Interest:** The authors declare no conflicts of interest.

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
