# Peer review of "Cooperative Learning Promoting Cultural Diversity and Individual Accountability: A Systematic Review"

_education, doi:10.3390/educsci14060567_

Round 1

Reviewer 1 Report

Comments and Suggestions for Authors

Cooperative learning promoting cultural diversity and individual accountability: a systematic review

General Comments for the Authors:

The authors present interesting research on the effect of cooperative learning. More specifically, if cooperative learning is beneficial at micro and macro levels, this should be implemented throughout the PE-lessons everywhere.

In general, the idea is clear, and the research method fits. Fairly enough, this is informative for the readers. However, if this paper is directed towards the results of cooperative learning, the literature review should present the working mechanism behind the expected relationships; the results must be more concise, and the conclusion more in line with the results.

Specific Comments:

Introduction:

The purpose of this study is presented clearly and is meaningful. However, the linking with the sustainable development goals is not very clear. Please expand on the rationale behind this relationship. Which of the social cognitive physical aspects are connected to these goals and why is CL effective on these aspects? For example, why should CL lead to cultural integration? Also, a clear definition and description of CL are missing.

Although I do like bold statements, this one seems to need a citation to prove it:

“CL rules under the principles that operate under the tenet of leave-no-one-behind promoting cultural diversity through equity and inclusivity while developing competences for sustainable development” (line 45).

Based on the reviews of Bores-García et al (2021) & Zhou et al. (2023), it is a strong argument to look more into the quantitative aspects of the impact.

Methodological:

The visual chart gives a good overview of the used process. However, we would suggest explaining a bit more about the methodological inclusion criteria. For example, is there a minimum intervention duration, use of instruments, etc.? Although some information is given (“non-quantitative”), it is unclear what the exact criterion behind this decision is. This is also a merit in the results section.

Results:

The results section gives an interesting insight into the different studies and is well-structured. As mentioned in the introduction, I did expect more information on the quantitative aspects of each study (see PRISMA guidelines item 19 “For all outcomes, present, for each study: (a) summary statistics for each group (where appropriate) and (b) an effect estimate and its precision (e.g. confidence/credible interval), ideally using structured tables or plots.” If possible, an effect size plot and some background information about the duration and content of the interventions could enrich this article. Remarkably, seven factors are mentioned in table 4, but in the methods section only four outcome measures are used as inclusion criteria “and outcomes (e.g., cognitive, social, affective, motor)”. It would be wise to either add this in the method section or reflect on this outcome in the discussion. Some bald statements, e.g., line 233 “demonstrates effectiveness”, aren’t based on rigorous studies and should start with the author to present the studies more objectively.

Discussion:

With respect to the content of the discussion, it’s recommended to add some points and to tone down the conclusion. Structure: consider reflecting on the impact of CL on the learning outcomes to make a stronger connection with the goal of the study.

Flashpoints and limitations:

The addition of personalized education is a bit off-topic. Considering CL to be a group-centered approach, one should expect to reflect on group and cooperation skills. Overall, more attention could be given to the limitations of this study. In our opinion, the conclusion ‘CL demonstrates significant positive impacts in PE, effectively improving students’ cognitive development and academic performance while increasing enthusiasm for physical participation. In comparison to traditional educational models, CL efficiently eliminates negative emotions and enhances students’ critical thinking’ is too strong. Firstly, this study is based on a mixture of studies which have different outcomes for different intervention. Although the studies (24 & 84) compare the traditional instruction methods with CL, the traditional method is completely different. Therefore, “in comparison to traditional educational models” should be left out.

Overall, this study holds an important topic, and we do hope that the authors initiate more research on Cooperative Learning.

Small remarks:

Line 89: some studies, please be concise.

Line 189: with noticeable impacts”, what is noticeable.

Line 379: some students, please be concise.

Line 415: some studies, which?

Author Response

Dear reviewer,

thank you very much for your valuable comments. We have answer one by one all the points concerning the revision. The answers are in the document.

Prof. Jordi Colomer

University of Girona

Spain

Reviewer 2 Report

Comments and Suggestions for Authors

Dear authors,

Thank you for the opportunity of reviewing the article entitled: " Cooperative learning promoting cultural diversity and individual accountability: a systematic review”. The article aims to analyse the effects of cooperative learning in physical education through other studies on the subject. In other words, it consists of a systematic review of the literature on the topic in question. Congratulations for such an interesting article. However, although it is a topic of great interest, it needs improvement and thorough revision. I hope you find my comments useful. Best wishes to the authors. Here are listed some comments.

Introduction: The introduction seems complete and rigorous, clarifying the most relevant concepts of the study by making references to recent scientific literature. In addition, it underlines the reasons and the need to carry out this review on cooperative learning.

To facilitate reading, especially in the headings, it would be better not to use acronyms (CL or PE).

Method:  Some details that would need justification in the text relate to some decisions on revision:

1. Why were only empirical studies selected as indicated in the inclusion criteria? What do you mean by empirical studies: those that are not theoretical or review studies or those that are only quantitative in nature? If the latter, why select only quantitative studies? You should justify this. Also, if so, there is a certain contradiction when, in line 141, you explain the PICOS method, and you do mention qualitative and mixed studies. You should clarify this.

2. Why did the selected articles have to be in English only? I understand that this is due to linguistic limitations, or is there another reason?

The Data analysis section is actually a data selection/collection section, at no point does it explain how the selected papers are analysed. Which brings me to my next question: how were the texts finally selected and from which the results are extracted analysed? This requires special attention: narrative analysis, thematic analysis, content analysis, ...? I insist, this is key as it gives meaning and rigorousness to the following sections.

It would also be necessary to include a section on the method of limitations. Although you subtly mention it at the end of the article, you should go into a little more depth, because in this type of study, showing the weakness gives, in a way, rigour to the process.

Results: I will briefly comment on the results which, without a context on the analysis processes developed, are difficult to judge.

A priori it seems to be a descriptive analysis of the content on aspects such as geographical distribution, characteristics of the samples analysed (educational stages), research objectives and results. As a synthetic description of the studies on what is being studied, how it has been carried out and what the main contributions are, it is correct. Perhaps you could even introduce some data on the main approaches of the studies (what type of methodological design, instruments used, sample size, etc.).

However, this being the case, I suggest you change your approach and objective included in the introduction (lines 49 onwards). For with these results and this process of analysis it is not possible to “quantitatively examine the impact of CL on students' multiple learning outcomes (motor skills, social skills, motivation for self-directed learning, affective experiences, etc.), nor did they analyze how the CL was applied in diverse age groups and multicultural contexts.” It simply describes the study area, highlighting the characteristics and main results. Therefore, far from “examines the effects of using cooperative learning (CL) in physical education (PE) at both micro and macro levels” as indicated in the abstract.

Therefore, I suggest nuancing the objectives so that they are in line with the research process and the results described.

Conclusions: In this section it would be very interesting to include a paragraph on the implications for practice of the results of this study.

So, as you will read, I have encountered a few problems that I feel need to be clarified for possible publication. I hope you find my comments useful.

Regards.

Author Response

Dear reviewer,

thank you very much for your valuable comments. We have answer one by one all the points concerning the revision. The answers are in the document.

Round 2

Reviewer 2 Report

Comments and Suggestions for Authors

Dear authors, 

After reviewing the changes made and the authors' responses to my comments, I feel that all the concerns highlighted in my first review have been addressed. Therefore, I now note that the article is now an improved version of the first submission in which the problems noted have been ironed out.

Congratulations. 

Yours sincerely.